# Application of Nanosilicon to the Sintering of Mg-Mg_2_Si Interpenetrating Phases Composite

**DOI:** 10.3390/ma14237114

**Published:** 2021-11-23

**Authors:** Anita Olszówka-Myalska, Hanna Myalska, Patryk Wrześniowski, Jacek Chrapoński, Grzegorz Cios

**Affiliations:** 1Faculty of Materials Engineering, Silesian University of Technology, Krasińskiego 8 Street, 40-019 Katowice, Poland; hanna.myalska@polsl.pl (H.M.); Patryk.Wrzesniowski@polsl.pl (P.W.); jacek.chraponski@polsl.pl (J.C.); 2Academic Centre for Materials and Nanotechnology, AGH University of Science and Technology, Al. A. Mickiewicza 30, 30-059 Kraków, Poland; ciosu@agh.edu.pl

**Keywords:** nanosized Si, Mg_2_Si, magnesium matrix composite, in situ composite, interpenetrating phases composite

## Abstract

The new in situ fabrication process for Mg-Mg_2_Si composites composed of interpenetrating metal/intermetallic phases via powder metallurgy was characterized. To obtain the designed composite microstructure, variable nanosilicon ((n)Si) (i.e., 2, 4, and 6 vol.% (n)Si) concentrations were mixed with magnesium powders. The mixture was ordered using a sonic method. The powder mixture morphologies were characterized using scanning electron microscopy (SEM), and heating and cooling-induced thermal effects were characterized using differential scanning calorimetry (DSC). Composite sinters were fabricated by hot-pressing the powders under a vacuum of 2.8 Pa. Shifts in the sintering temperature resulted in two observable microstructures: (1) the presence of Mg_2_Si and MgO intermetallic phases in α-Mg (580 °C); and (2) Mg_2_Si intermetallic phases in the α-Mg matrix enriched with bands of refined MgO (640 °C). Materials were characterized by light microscopy (LM) with quantitative metallography, X-ray diffraction (XRD), open porosity measurements, hardness testing, microhardness testing, and nanoindentation. The results revealed that (n)Si in applied sintering conditions ensured the formation of globular and very fine Mg_2_Si particles. The particles bonded with each other to form an intermetallic network. The volume fraction of this network increased with (n)Si concentration but was dependent on sintering temperature. Increasing sintering temperature intensified magnesium vaporization, affecting the composite formation mechanism and increasing the volume fraction of silicide.

## 1. Introduction

Metal matrix composite materials have been intensely examined over the past few decades. New materials and technologies are constantly being investigated to obtain more desirable material properties than those of conventional materials. In recent years, the main development trends have lain in the application of nanosized functional phases, in situ phase formation, as well as interpenetrating phase system processing.

The use of nanosized components, mostly ceramic (e.g., SiC [1,2,3,4], SiO_2_ [5], AlN [6], Al_2_O_3_ [5,7,8,9], B_4_C [10,11], TiB_2_ [12], TiC [13], or Sm_2_O_3_ [14]) or carbon (e.g., fullerene, nanotubes, and graphene [15,16,17,18]), allows for the implementation of Orowan’s mechanism and grain size refinement within the metal matrix. Combining these concepts can be used to improve the mechanical properties and wear resistance, in addition to changing the friction coefficient. The primary problem with this process is the agglomeration of nanosized raw components prior to or during the consolidation process with liquid or powder metals.

In situ metal matrix composites are the next research trend, as these materials are characterized by stronger matrix bonding and allow for improved control over new phase formation in terms of size, morphology, and distribution [19,20]. Expected structural effects can be tailored through the reinforcement or addition of a functional phase via chemical reactions between the metallic matrix and specific solid or gaseous additions [19,21,22,23] or directional crystallization of eutectic alloys [24,25]. In the literature, various precursor materials are used for the in situ formation of reinforcing particles. For instance, TiB can form the TiB_2_ phase in Al matrices [26,27], SiO_2_ or Si can form the Mg_2_Si phase in Mg matrices [20,28,29,30], and TiB and TiC phases can be formed from B_4_C [31,32] (or TiC from carbon precursors [33]) in Ti matrices. Processing is typically completed by stirring a liquid matrix (stir casting) [30,34,35] or by powder metallurgy [22,28]. 

Interpenetrating phase composites (IPCs) are composed of two or more phases that are topologically continuous and fully interconnected throughout the microstructure. IPCs have been formed using a wide variety of technological procedures. Two primary fabrication methods for ceramic/metal systems have been reported in the literature—(1) interpenetrating structures formed from powder mixtures of fixed phase composition or the (2) filling of open-celled ceramic foams with a metallic component. An example of the first fabrication method is the Al_2_O_3_/Ni system where studies have reported the sintering of a compacted powder mixture to form an interpenetration structure using a specific component granulation and volume fraction [36,37]. Similar effects were obtained when the powder mixture was applied by slip casting [38]. The primary problem for all systems (i.e., ceramic–ceramic, metal–metal, and ceramic–metal) using these methods is the proper selection of sintering conditions to ensure low porosity and strong bonding between powder grains. Open-celled foam or other open-network substrates have been filled by liquid metal using pressure infiltration [39,40,41,42,43] and centrifugal casting [44] or by the vibro-compaction of metal powder-foam systems and additional sintering under pressure [45]. The main technological challenges for these technologies are ensuring strong bonding between components and low composite porosity. Additionally, designing interpenetrating composite structures faces technological limitations such as the fabrication of foams with a required cell geometry and liquid- or solid-component penetration conditions.

The aim of this experimental work was to reveal the step-by-step formation mechanism of an interpenetrating in situ composite using microsized magnesium and nanosized silicon ((n)Si) powder. These experiments included the preparation of ordered Mg-(n)Si mixtures via the sonic method and subsequent composite fabrication. The mixture morphology was characterized for variable (n)Si volume fraction using SEM, and the thermal analysis of Mg-(n)Si-mixtures was measured using DSC. The Mg-Mg_2_Si composites sintered at temperatures below the melting point (580 °C) and at the melting point (640 °C) were examined and characterized by LM, SEM, quantitative metallography, open porosity measurements, hardness testing, and nanoindentation. These results showed the in situ formation of Mg_2_Si in the α-Mg matrix and revealed the role of (n)Si content in the ordered nano/micro-powder mixture and sintering temperature in the composite morphology design.

## 2. Materials and Methods

Microsized Mg powder (25–66 μm, 99% purity, Aldrich) and (n)Si (<100 nm, 98% purity, Aldrich) were used as precursor materials for composite manufacturing. To prepare Mg/Si powder mixtures, a sonic method (45 kHz) was used to fabricate mixtures containing 2–6 vol.% (n)Si. The mixing procedure was similar for all Mg-(n)Si powder mixtures. First, the (n)Si powder was ultrasonically mixed with ethanol for 1 h to separate the nanoparticles. Then, Mg powder was gradually added to the continuously ultrasonically mixed (n)Si-alcohol suspension. Ultrasonication was continued until the mixture was transparent. Finally, the alcohol was removed and the resultant powder mixture dried. The same procedure was applied for Mg-Si mixtures containing 6 vol.% of microsized Si powder (325 mesh, 99% purity, Aldrich) to be used as reference material for thermal analysis.

To characterize the synthesized initial powder mixtures, a field emission gun scanning electron microscope (FE-SEM Hitachi-4200S, Hitachi, Japan) was used. 

To measure thermal effects in the Mg-Si system, differential scanning calorimetry (DSC) analysis was performed under an argon atmosphere (99.9999%, 20 mL/min) using a Pegasus 404 F1, Netzsch apparatus. A heating and cooling rate of 5 °C/min was used for the 6 vol.% nanosized Si in the microsized Mg sample, and a heating and cooling rate of 10 °C/min was used for the 6 vol.% microsized Si in the microsized Mg sample. A temperature range from 30 to 700 °C was used.

Hot pressing under a vacuum of 2.8 Pa was performed using a Degussa press for composite fabrication. During the process, powder mixtures with 2, 4, and 6 vol.%. contents of (n)Si were annealed at 150 °C for 1 h before compaction under 1.5 MPa at 300 °C for 10 min. Finally, the samples were heated up to 580 °C under 8 MPa and held for 5 min before cooling. For composite sample comparison, a pure magnesium powder reference sample was synthesized under the same conditions. 

To show the role of sintering temperature in interpenetrating phase formation, Mg-Mg_2_Si composites were synthesized using the same Mg-(n)Si mixtures, compaction time and pressure, and cooling conditions, but the sintering temperature was increased from 580 °C to the melting point temperature, 640 °C. The powder mixtures were labeled Mg-2(n)Si, Mg-4(n)Si, and Mg-6(n)Si, and the obtained sinters were labeled (including composition and sintering temperature) Mg-2(n)Si/580, Mg-4(n)S/580, Mg-6(n)Si/580, Mg-2(n)Si/640, Mg-4(n)S/640, and Mg-6(n)Si/640.

To confirm the phase composition, X-ray diffraction (XRD) (JDX-7S diffractometer) and the JCPDS-International Centre for Diffraction Data were used (Mg 35-0821, Mg_2_Si 75-0445, MgO 87-0653). The composites’ densities and porosities were measured using the Archimedes method according to the standard no. PN-EN 993-1:2019-01.

The microstructure of polished composite samples was examined by the light method (LM, Eclipse MA200 Nikon, Tokyo, Japan) where the Mg_2_Si phase was easily distinguishable due to its characteristic blue color. The quantitative metallography was employed (Met-Ilo software by Janusz Szala, Poland) to quantitatively describe microstructural changes as a result of increasing (n)Si content in the initial powder mixture and sintering conditions. The Mg_2_Si phase and obtained cell structure were characterized without etching. 

The composites’ hardness was determined using the Vickers method (Zwick 110 hardness tester, Ulm, Germany) under a load of 3 kg (HV3) with a 10 s dwell time, and at least 30 indentations were conducted on each sample. The microhardness HV0.2 (load of 0.2 kg, dwell time 10 s) was measured in different areas of the formed composite such as the magnesium-based cells’ interior, silicide-based skeleton, and large compact Mg_2_Si microareas. At least 30 indentations were conducted on each structural element type. 

The same microareas of 580 series sinters (Mg-2(n)Si/580, Mg-4(n)Si/580, and Mg-6(n)Si/580) were examined via nanoindentation. The measurements were carried out using the G200 nanoindenter equipped with an XP head (KLA-Tencor, Milpitas, California, USA) on polished cross-sections. A Vickers diamond tip (Synton-MDP, Nidau, Switzerland) was used with a 1 mN maximum load, 15 s was used for loading segments, and 10 s of hold time was used at maximum load. The indenter tip area function was calibrated using a fused silica standard sample. The hardness and Young’s modulus were calculated using the Oliver–Pharr method. The maximum allowable drift before measurement was set to 0.050 nm/s, and drift correction after measurements was performed. For the Young’s modulus calculation, a Poisson’s ratio of 0.3 was assumed. Each sample was tested 10 times.

## 3. Results and Discussion

### 3.1. Formation of the Ordered Mg-(n)Si Powder Mixture

SEM was used to characterize the morphology of the powder mixtures prior to sintering. The observed powder mixture morphologies are presented in Figure 1, Figure 2 and Figure 3.

SEM imaging of the (n)Si-Mg powder mixtures containing variable (n)Si concentrations (Figure 1, Figure 2 and Figure 3) showed adhesion between the (n)Si particles and microsized Mg grains after sonication. It must be mentioned that, in the mixture containing 2 vol.% of (n)Si, only the separated Mg particles were visible at the macroscopic level, although a beige color was observable compared with the initial grey color of the pure Mg powder. This color change was produced by the presence of (n)Si particles or nanoagglomerates on the Mg surface, which were visible under SEM (Figure 1). Increasing the (n)Si content to 4 vol.% produced a discontinuous (n)Si layer on the Mg surface (Figure 2). Additionally, some of the microsized Mg particles were connected by bridges formed from agglomerated (n)Si grains, although this was not observable at the macroscopic scale. As the (n)Si concentration was increased to 6 vol.%, a continuous (n)Si layer was formed (Figure 3). Additionally, the size and frequency of (n)Si bridges increased. At the macroscale, this mixture was slightly clumped after drying.

The analysis of structural effects produced by ultrasonic mixing confirmed that the applied method ensures good adhesive contact between the deagglomerated nanopowder and microsized powder. Moreover, the microstructure of the obtained powder mixtures was dependent on (n)Si concentration, where concentration increases led to the formation and development of (n)Si bridges within the Mg matrix. This type of (n)Si secondary agglomeration can form microsized silicides in the magnesium matrix under pressurized sintering. However, it can be assumed that an increase in the nanocomponent content in the nano/micro powder mixture is achievable without producing this bridging effect by using a finer granulation of the microsized component and subsequently increasing the specific surface area. Current efforts have been dedicated to this progress. Generally, the phenomenon of ordered mixture formation consists of two main steps: uniform deposition of individual nanoparticles in a microsized powder and secondary agglomeration of nanoparticles that produces effects such as continuous layer and bridge formation of the nanocomponent (Figure 4).

### 3.2. Thermodynamic Aspects of Interaction in Mg-Si System

The theoretical analysis of chemical interactions between Mg powder and Si was performed to include the presence of MgO at the magnesium surface and strong affinity of Si to oxygen (SiO_2_). Thermodynamic data of these interactions were calculated for different temperatures using Chemistry program HSC 6.2, and the results are summarized in Table 1. Based on the thermodynamic data, magnesium silicide (Mg_2_Si) is a product in the examined Mg-Si system, and MgO is unreactive with silicon. Moreover, the silicon oxide theoretically present in trace amounts in pure silicon powder is strongly unstable in the presence of magnesium and transforms to MgO and Mg_2_Si or pure Si. This means that, after sintering the powder mixtures, we can expect Mg, Mg_2_Si, and MgO phases only.

To investigate thermal effects in the system where nanocomponents were deposited on the microcomponent by ultrasonication, DSC analysis was performed. The two mixtures containing different sizes of Si particles (6 vol.% of (n)Si and microsized Si in microsized Mg) were examined for comparison. Both mixtures were prepared via ultrasonication, but, unlike mixtures containing (n)Si, the SEM examinations of the microsized Si-Mg powder mixture did not reveal any specific grain ordering effects and only confirmed the uniform distribution of the silicon particles. 

Examples of DSC curves obtained for (n)Si-Mg and microsized Si-Mg powder mixtures are presented in Figure 5. The thermal effect values measured for all four experimental variants are shown in Table 2. Knowledge of those effects was crucial in determining the influence of silicon powder granulation and mixture ordering on silicide formation, as well as the selection of sintering parameters.

An exothermic reaction between Mg and Si was observed in the heated powder mixtures before the endothermic peak associated with the Mg melting point. Heating rate effects were observed, although these effects were independent of the Si grain size. Faster heating resulted in shifts in the reaction peak to higher temperature and more symmetric data. In the case of (n)Si, the initiation and maximum reaction temperatures were lower (approx. 50 K) compared with the microsized Si-Mg mixture (Table 2). The observed effect of starting a reaction earlier may be due to two reasons. One of them is the known fact of a more developed specific surface of nanopowder, and the other is an ordered morphology of the mixture of nano- and micro-powders due to the ultrasonic mixing. The nanopowder forms a compact coating on the micropowder and the initiation of the reaction occurs earlier. Additionally, symmetry was not observed in the exothermic peak (Figure 5a), and it was wider after achieving a maximum value. 

The thermal parameter values measured for melting and solidification showed that the onset and peak maximum temperature were a few Kelvins less in the case of (n)Si, requiring further study. Results measured for the microsized powder mixture are in good agreement with the experimentally measured eutectic temperature reported in the literature [46,47].

### 3.3. Characteristics of Composites Sintered at 580 °C

XRD analysis (Figure 6) confirmed the presence of α-Mg and MgO from the pure magnesium powder and formation of Mg_2_Si silicide as a reaction product of magnesium and silicon. These results were in agreement with the theoretical data presented in Table 1.

Composite microstructure observations for the 2, 4, and 6 vol.% (n)Si in Mg mixtures sintered at 580 °C are presented in Figure 7, Figure 8 and Figure 9. A cellular structure with a multiphase network formed by the mixture of very fine Mg_2_Si (blue) and MgO (beige) particles in the α-Mg matrix was visible. The silicide was formed in situ, while the oxide originated from the magnesium powder surface was fragmented during reactive sintering. The cell shapes were similar in size to the initial magnesium powder grains (Figure 7a, Figure 8a, and Figure 9b), suggesting the importance of the chosen microsized powder granulation in the final design of the intermetallic/ceramic morphology. With increasing (n)Si, the skeleton was more distinct and its volume increased, and ~10 µm pure silicide agglomerates were seen (Figure 8c). The agglomerates grew more frequent with increasing (n)Si content and contained small additions of submicrosized MgO particles. This type of microstructure suggests their origin from the bridges or agglomerates formed during the preparation of Mg-(nSi) mixtures. Another structural effect was observed at the cross-sections in magnesium-rich areas that were closed in Mg_2_Si based where single globular Mg_2_Si inclusions ~1 µm were present (Figure 8c and Figure 9c).

The observed structural phenomena were related to the heating temperature, which was above the temperature for Mg_2_Si formation and below the melting point (Figure 4 and Table 2). At first, the silicon nanoparticles formed submicrosized and finer Mg_2_Si particles that could agglomerate. At the same time, the oxide film fragmented (Figure 7b and Figure 8b). These processes occurred near the magnesium powder grain surface. Moreover, due to local temperature increases by strong exothermal reactions, the eutectic temperature was exceeded, allowing for silicide dissolution and further precipitation. Additionally, previously formed Mg_2_Si from (n)Si could relocate in liquid magnesium.

For quantitative image analyses of the sintered composites, detection, automatic correction, and manual correction were used to examine the microstructural elements:In situ-formed Mg_2_Si phase to examine the effects of the Mg-(n)Si reaction (500× magnification).Inner α-Mg-based region closed by in situ-formed skeleton to characterize the interpenetrating system (200× magnification).

Binary images of these elements after detection, automatic correction, and manual correction are presented in Figure 10.

Quantitative image analysis of the Mg_2_Si phase formed in situ during composite sintering at 580 °C is presented in Table 3. With increasing (n)Si volume fraction, both the average surface and volume fraction of silicide particles increased, while the shape factor remained similar. The analysis of cross-sectioned particles divided into five size classes showed a similar number of particles under 0.1 μm^2^ with the majority of particles falling in the ranges of 0.1–1 and 1–10 μm^2^, independent of mixture composition. That quantity of Mg_2_Si particles over 100 μm^2^ increased with increasing (n)Si due to differences in the ordered powder mixture morphology produced by ultrasonication. The nearly flat Mg-(n)Si reaction front at the Mg powder surface induced the formation of micron-sized Mg_2_Si particles or smaller, but the (n)Si bridges and secondary agglomerates (Figure 2a and Figure 3a) transformed during sintering to form larger, more numerous intermetallic particles within the formed composites.

The results of quantitative metallography used to analyze the mechanism of interpenetrating phase formation are presented in Table 4. A decrease in α-Mg rich areas doped with Mg_2_Si and surrounded by a multiphase skeleton for increasing (n)Si content in the initial powder mixture was revealed as Mg was consumed to form Mg_2_Si (Table 4).

To describe the material property changes induced by (n)Si application in Mg-based sintering, open porosity and different hardness measurement techniques such as HV3, microhardness HV0.2, and nanoindentation (Table 5 and Table 6) were applied. HV0.2 and nanoindentation measurements of the silicide-based skeleton and Mg-based cell interior were taken into consideration.

In general, increasing (n)Si concentration in the initial powder mixture increased the HV3 hardness in Mg-based sintered composites. In the case of Mg-6(n)Si/580, the hardness was 50% higher than that of the sintered pure Mg, which served as a reference sample. The improvement of hardness was due to silicide formation and reduced porosity compared with sintered pure Mg. HV0.2 measurements focused on the grain center and surrounding secondary phases differed depending on the powder mixture composition. The microhardness of the composite cell interior was higher than that of the Mg in the reference sinter, and increased with (n)Si content. The skeletons were characterized by a microhardness twice as high as the cells’ interiors. For the Mg-4(n)Si and Mg-6(n)Si samples, the skeleton microhardness was similar and approximately 30% higher than that of the Mg-2(n)Si sample. Nanoindentation confirmed a higher hardness of the skeleton compared with the α-Mg matrix. The matrix hardness increased with increasing (n)Si content. It must be mentioned that the nanohardness of the reference sinter in the oxide regions located at the previous powder grains was similar to the skeleton, suggesting that the MgO film was very compact and strongly bonded with Mg. The highest hardness and Young’s modulus in the nanohardness measurements were in compact Mg_2_Si particles, which were ~10 µm in size. Similar values can be expected in the continuous silicide network with additional increases in (n)Si in the initial powder mixture.

Hardness changes in the Mg-Mg_2_Si interpenetrating phase composites were in good agreement with the presented microstructure examinations. These shifts can be explained by secondary phase growth and by structural effects in the α-Mg grains, the second main interpenetrating element. In Mg, grain inclusions of fine Mg_2_Si particles were observed, and that effect suggests changes in grain size dependent on (n)Si concentration in the raw powder mixture. This issue is examined further. 

The force–depth curves obtained from nanoindentation (Figure 11) clearly exhibited differences in plastic deformation for the same applied force on different microstructural elements, confirming higher stiffness of the multiphase skeleton compared with the α-Mg matrix. The highest Young’s modulus was observed for Mg_2_Si agglomerates. The curves obtained for the skeleton region became more similar to the curves for Mg_2_Si agglomerates with increasing (n)Si.

The measurement of HV3 hardness allowed the determination of general information about the properties of the obtained materials and the influence of technological parameters such as the volume fraction of (n)Si and the sintering temperature. The characteristics of HV0.2 microhardness and nanohardness of the skeleton and cell interior gave similar information on the influence of technological parameters, and the obtained results were consistent with the direction of changes in the hardness of the composite. 

### 3.4. Characteristics of Composites Sintered at 640 °C

The microstructure of longitudinally cross-sectioned sinters obtained from powder mixtures of the same composition and ultrasonicated but sintered at 640 °C is shown in Figure 12, Figure 13 and Figure 14. Table 7 shows the characterization results of Mg_2_Si, composite porosity, and HV3 measurements. A few characteristic effects were observable. Semispherical Mg_2_Si particles (blue) (~1–4 µm in size) formed around α-Mg grains (white) enriched with fine MgO particles (beige). 

This was the structure of interpenetrating phases although different in comparison to the morphology seen in the composites sintered at 580 °C. Moreover, the volume fraction of the Mg_2_Si phase in the composites was several dozen times higher compared to the (n)Si volume fraction in powder mixtures applied in experiments, and that effect was significantly more intense as with the composite sintered at 580 °C. Similar to the latter, increasing (n)Si content increased the Mg_2_Si phase thickness and irregularity. The next structural difference was regarding the MgO film behavior. Specifically, the oxide contour of previous Mg powder grains and fragmented oxide flakes was absent. Instead, a regular distribution of submicrosized or nanosized MgO particles was observed in the form of bands in the α-Mg matrix or within the secondary Mg_2_Si phase. Additionally, increasing (n)Si concentration in composites sintered at 640 °C decreased the porosity and increased hardness HV3. Although this trend was also observed in the composites sintered at 580 °C, the effects were magnified where the hardness was over two times higher and the porosity was similar or lower. This was due to the sintering temperature and DSC results shown in Figure 5. Heating (n)Si-Mg powder mixtures at 640 °C in a Degussa press produced a sharp rise in temperature due to the exothermic transformation of (n)Si into Mg_2_Si. This reaction has even been termed as “explosive” [48] and explains the decomposition of the brittle MgO film to form nanosized particles. Moreover, local temperature increases resulted in intense Mg evaporation and a strong loss in the final volume fraction of the synthesized composite. Generally, it is expected that high-temperature self-propagating synthesis (SHS) occurred during Mg-Mg_2_Si composite processing. This also explains why the (n)Si agglomerates in nano/micro-powder mixtures (Figure 1, Figure 2 and Figure 3) did not produce significant disordering in the interpenetrating phases’ composite formation seen (Figure 8c) in composites sintered at 580 °C.

These results led to the proposal of the transformation model of ordered nanosized/microsized reactive powder mixtures formed by ultrasonication into interpenetrating phase composites (Figure 15). Two possible sintering temperature-dependent methods of interpenetrating phase composite formation are shown. 

The first method occurs due to (n)Si interactions with Mg at lower temperatures where secondary phase growth is determined by the location of the oxide film on the raw metal powder where (n)Si is cumulated. The skeleton is formed by the mixture of Mg_2_Si, Mgα, and MgO flakes. The second mechanism is due to a more dynamic reaction where the skeleton is formed only by microsized globular Mg_2_Si particles, and the geometry is not determined by the geometry of the initial microsized metal powder. The nanosized MgO particles coming from the refined film form bands in α-Mg and are also detected in Mg_2_Si regions. It should also be noted that, when a higher processing temperature is applied, the intermetallic phase network is thicker and metal vaporization significantly increases, resulting in a higher silicide volume fraction.

## 4. Conclusions

In this paper, a powder mixture consisting of a nanosized component reacted with a microsized metal matrix was successfully proposed to obtain a composite with an interpenetrating network. These Mg-Mg_2_Si composites were fabricated by hot pressing under vacuum. The nanosized Si and microsized Mg were chosen to obtain ordered mixtures with three different nanocomponent compositions using ultrasonication. This fabricated mixture was examined by DSC to determine the reaction parameters. These results indicated two sintering temperatures for hot pressing under vacuum, between the reaction temperature and melting point or at the melting point. Sintering temperature-dependent changes were observed in the microstructure for ordered nano/micro-powder mixtures, revealing possibilities of composite materials’ fabrications with different interpenetrating phase morphologies. The detailed conclusions are as follows:Ultrasonic mixing of the nanosized powder with the microsized powder promoted adhesive bonding between particles, and a nanosized powder concentration-dependent ordered configuration was formed.Two effects were observed in the powder mixture: (1) single (n)Si particles were deposited on the microsized Mg surface to form a continuous nanolayer, and (2) bridges formed between the Mg particles or single secondary agglomerates, which consisted of adhesively connected (n)Si grains.Replacing the microsized Si particles with (n)Si particles in the microsized Mg powder base influenced the thermal effects, as measured by DSC. The initial and maximum peak temperature values decreased by approximately 50 K for exothermal Mg_2_Si phase formation.Mg-Mg_2_Si composites formed two types of interpenetrating network morphologies. These were produced by reactive synthesis of the ordered nano/micro-powder mixture and was dependent on sintering temperature. The main differences related to the arrangement of the Mg_2_Si phase and its volume fraction, as well MgO morphology changes.In powder mixtures heated to temperatures below the melting point, the submicrosized and globular Mg_2_Si particles formed a network around α-Mg microsized grains located in the region of the previous Mg powder surface and were enriched with the fragmented MgO film. This suggests the possibility of multiphase skeleton size design by raw Mg powder geometry. Increasing (n)Si concentration increased the volume fraction and thickness of the Mg_2_Si phase and increased the hardness and modulus of the interpenetrating phases.When sintering was performed at the melting point, the reaction grew more intense, and an interpenetrating network of globular microsized Mg_2_Si particles in α-Mg was formed. The MgO film was broken down into submicrosized and nanosized particles to create bands in α-Mg and was present in the Mg_2_Si skeleton. Moreover, intense Mg vaporization resulted in higher silicide concentrations in the synthesized composite and higher hardness compared to sintering at lower temperature.

## Figures and Tables

**Figure 1 materials-14-07114-f001:**
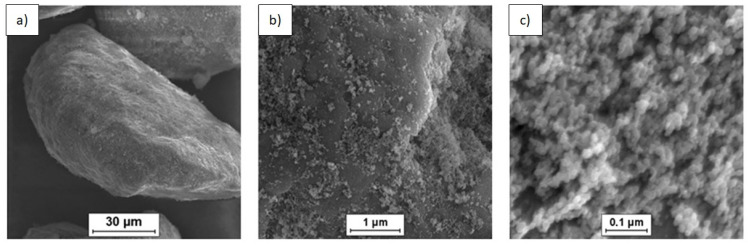
SEM micrographs of the Mg-2(n)Si powder mixture observed with different magnifications. Locally visible nanoagglomerates of (n)Si particles on the microsized Mg grains’ surface.

**Figure 2 materials-14-07114-f002:**
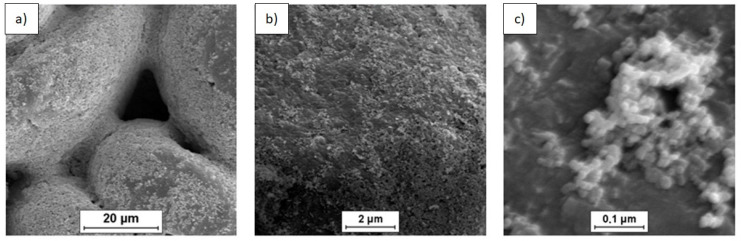
SEM micrographs of the Mg-4(n)Si powder mixture observed with different magnifications. Layers of (n)Si agglomerates are visible on the surface of the microsized Mg grains. The (n)Si agglomerates form bridges (Figure 2a).

**Figure 3 materials-14-07114-f003:**
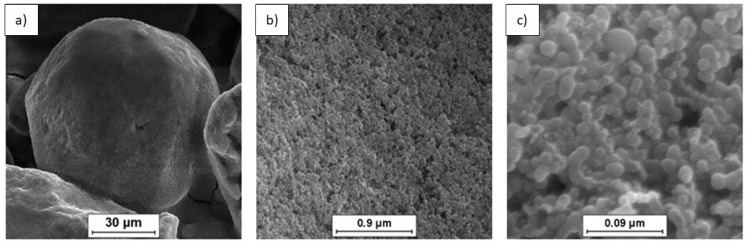
SEM micrographs of Mg-6(n)Si powder mixture observed with different magnifications. A continuous uniform layer of (n)Si deposited on the microsized Mg surface and bridges (Figure 3a) formed by (n)Si.

**Figure 4 materials-14-07114-f004:**
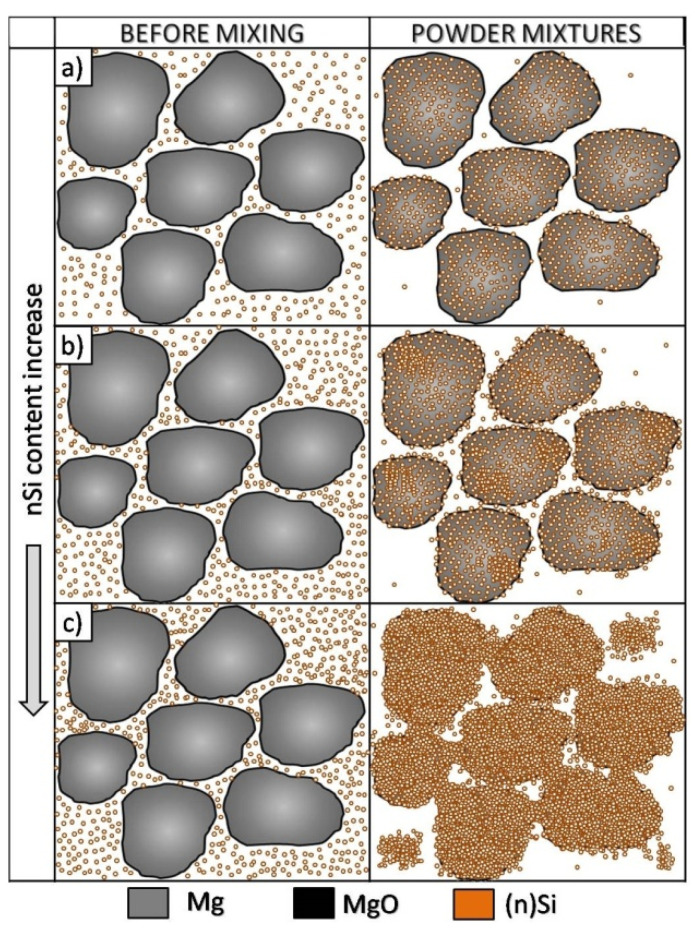
Scheme of ordered powder mixture morphology changes for the microsized/nanosized system for increasing nanocomponent concentration using ultrasonic mixing: (**a**) deposition of single nanoparticles on microsized component surface, (**b**) nanocoating formation by nanocomponent secondary agglomeration, (**c**) increase in nanocoating thickness and formation of bridges and secondary agglomerates.

**Figure 5 materials-14-07114-f005:**
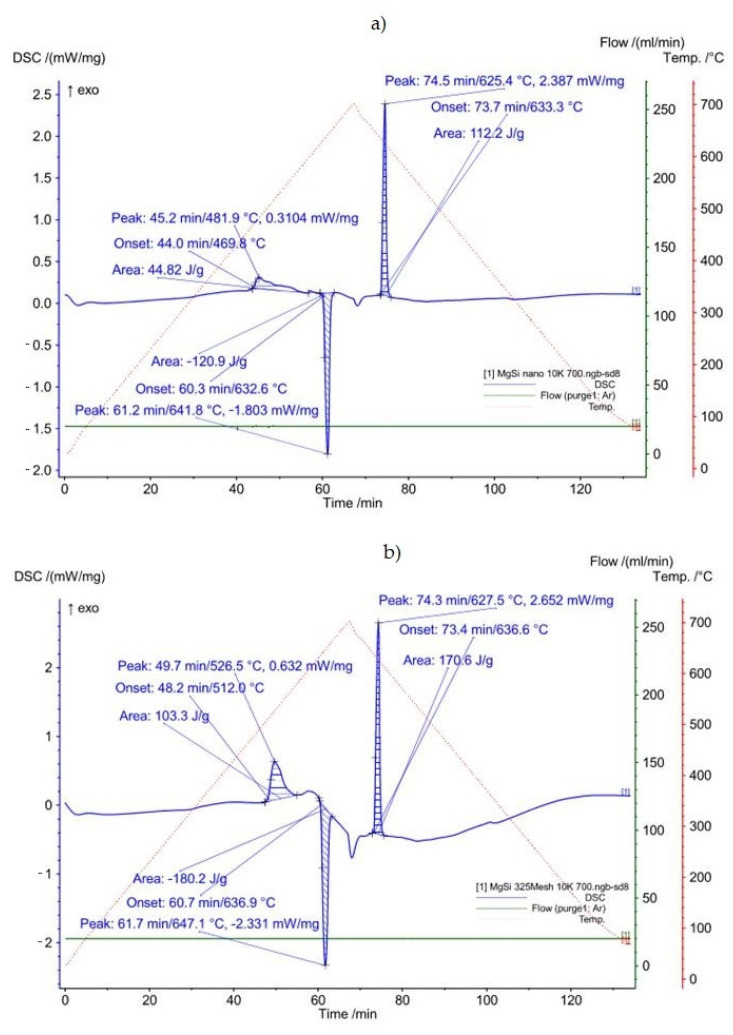
DSC profiles obtained for powder mixtures of (**a**) microsized Mg with 6 vol.% nanosized Si and (**b**) 6 vol.% microsized Si, heated and cooled at a rate of 10 K/min.

**Figure 6 materials-14-07114-f006:**
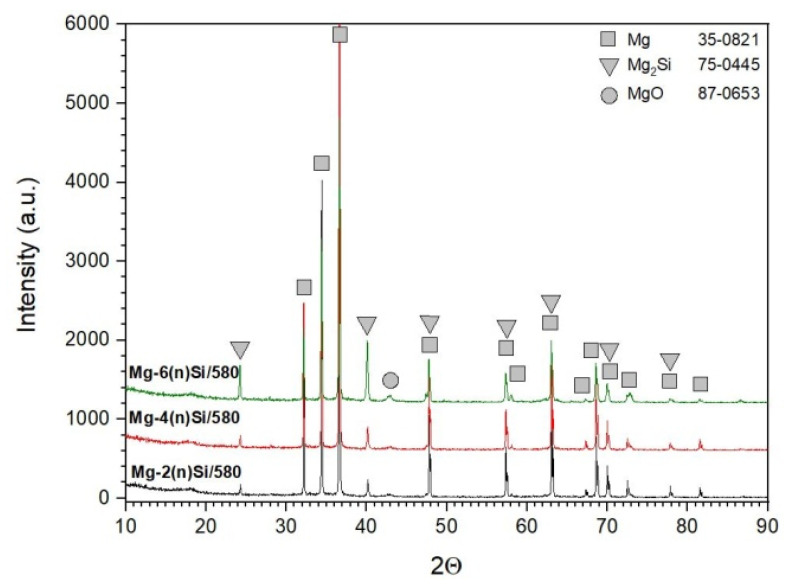
XRD patterns of Mg-Mg_2_Si composite obtained from 2, 4, and 6 vol.% (n)Si in Mg powder mixture.

**Figure 7 materials-14-07114-f007:**
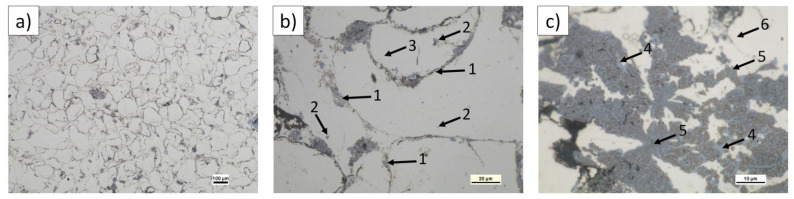
LM micrographs of Mg-2(n)Si/580 composite observed with different magnifications: (**a**) cellular microstructure with fine phases focused in the original magnesium powder grain region; (**b**) morphology of single cells with irregular skeleton of Mg_2_Si and fragmented MgO film (1), fine Mg_2_Si particles (2), located in α-Mg (3); and (**c**) Mg_2_Si silicide agglomerates (blue) (4) and MgO (beige) (5) in the α-Mg matrix (light) (6).

**Figure 8 materials-14-07114-f008:**
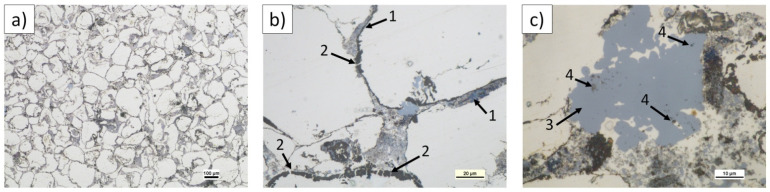
LM micrographs of Mg-4(n)Si/580 composite observed with different magnifications: (**a**) cellular microstructure with fine phases focused in the original magnesium powder grain region; (**b**) morphology of single cells with irregular skeleton of fine Mg_2_Si (1) and fragmented MgO film (2); and (**c**) Mg_2_Si silicide agglomerate (3) with single MgO particles (4) inside.

**Figure 9 materials-14-07114-f009:**
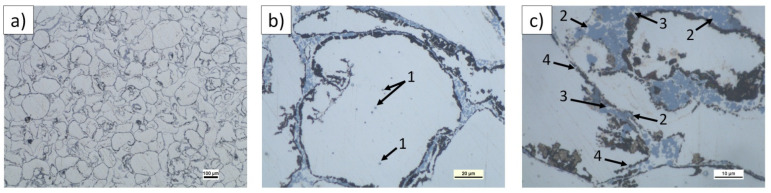
LM micrographs of Mg-6(n)Si/580 composite observed with different magnifications: (**a**) cellular microstructure with fine phases focused in the original magnesium powder grain region; (**b**) morphology of single cells with fine Mg_2_Si particles (1); and (**c**) morphology of multiphase skeleton-Mg_2_Si silicide (2) surrounded by α-Mg (3) and coated with fragmented MgO film (4).

**Figure 10 materials-14-07114-f010:**
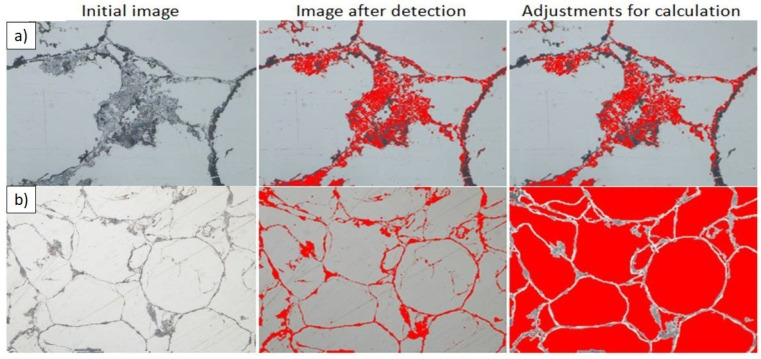
Examples of the detection procedure used for Mg-2(n)Si/580 composite characterization: (**a**) Mg_2_Si areas and (**b**) inner α-Mg-based regions of composite cells.

**Figure 11 materials-14-07114-f011:**
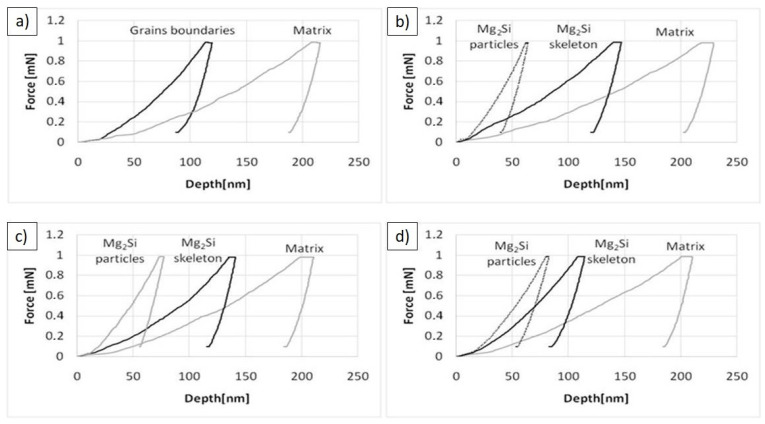
Nanoindentation force–depth curves determined for different elements of microstructure: (**a**) reference Mg/580 sinter, (**b**) Mg-2(n)Si/580 composite, (**c**) Mg-4(n)Si/580 composite, and (**d**) Mg-6(n)Si/580 composite.

**Figure 12 materials-14-07114-f012:**
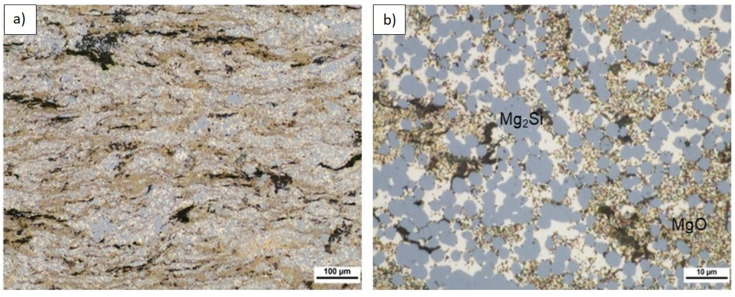
LM microstructure of Mg-2(n)Si/640 composite observed with different magnifications, Mg_2_Si skeleton (blue), α-Mg (light), submicrosized MgO (beige), and pores (black) formed by sintering or sample preparation.

**Figure 13 materials-14-07114-f013:**
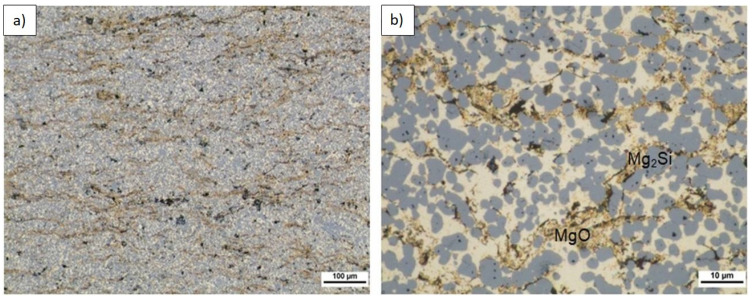
LM microstructure of Mg-4(n)Si/640 composite observed with different magnifications, Mg_2_Si skeleton (blue), α-Mg (light), submicrosized MgO (beige), and pores (black) formed by sintering or sample preparation.

**Figure 14 materials-14-07114-f014:**
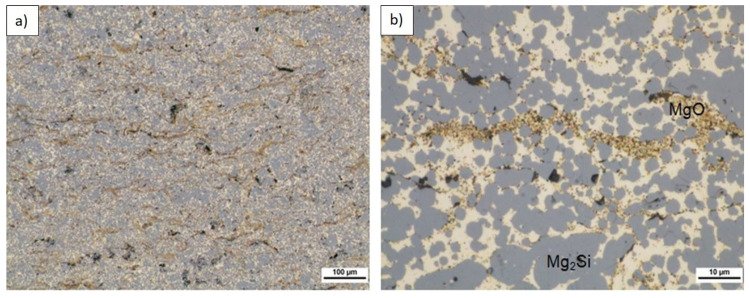
LM microstructure of Mg-6(n)Si/640 composite observed with different magnifications, Mg_2_Si skeleton (blue), α-Mg (light), submicrosized MgO (beige), and pores (black) formed by sintering or sample preparation.

**Figure 15 materials-14-07114-f015:**
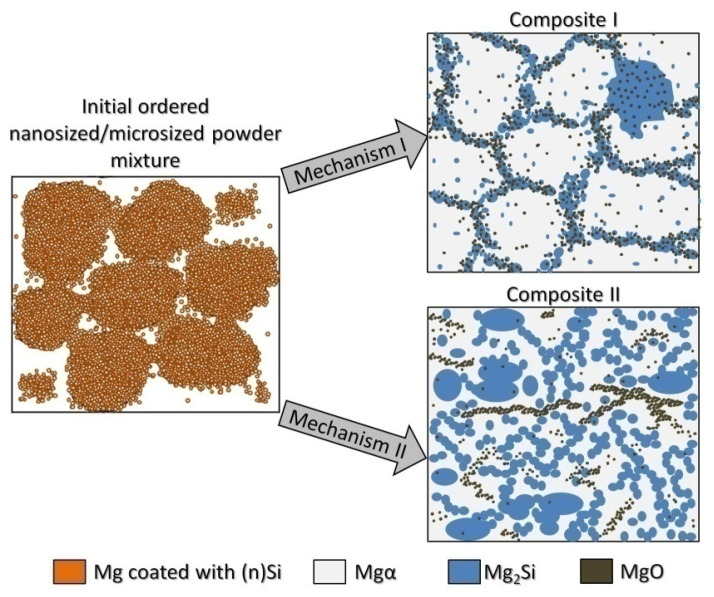
Structural model of interpenetrating phase Mg-Mg_2_Si composite formation from the ordered nanosized Si/microsized Mg powder mixture. Two possible microstructure types: I—in α-Mg matrix surrounded by skeleton of Mg_2_Si enriched with MgO and α-Mg where the cells’ geometry is determined by initial Mg powder granulation; II—in α-Mg-based matrix where the Mg_2_Si phase forms a skeleton with a geometry independent of the initial Mg powder granulation or MgO bands present in α-Mg.

**Table 1 materials-14-07114-t001:** Gibbs free energy for selected reactions in Mg-Si system at temperature range of 450–650 °C.

Reaction	Gibbs Potential ΔG, (kJ)				
	450 °C	500 °C	550 °C	600 °C	650 °C
2Mg + Si → Mg_2_Si	−71.7180	−71.2681	−70.8110	−70.3453	−69.8660
4Mg + SiO_2_ → Mg_2_Si + 2MgO	−339.7761	−337.6133	−335.4132	−333.1467	−330.8277
2Mg + SiO_2_ → 2MgO + Si	−268.0581	−266.3452	−264.6022	−262.8015	−260.9616
2MgO + Si → SiO_2_ + 2Mg	268.0581	266.3452	264.6022	262.8015	260.9616
MgO + SiO_2_ → MgSiO_3_	−37.6175	−37.8566	−38.0824	−38.2707	−38.4607
MgSiO_3_ + 4Mg → Mg_2_Si + 3MgO	−302.1586	−299.7566	−297.3308	−294.8760	−292.3669

**Table 2 materials-14-07114-t002:** Thermal effects measured for the Mg-Si system using DSC.

Mixture Composition	Heating Rate, (K/min)	Thermal Effects					
		Reaction 2Mg + Si → Mg_2_Si		Melting		Solidification	
		Onset, (°C)	Peak, (°C)	Onset, (°C)	Peak, (°C)	Onset, °C	Peak, (°C)
Mg + 6 vol.%(n)Si	5	455.4	464.3	633.4	640.9	633.6	627.0
	10	469.8	481.9	632.6	641.8	633.3	625.4
Mg + 6 vol.%Si 325 mesh	5	508.0	520.5	637.4	644.9	637.3	629.6
	10	512.0	526.5	636.9	647.1	636.6	627.5

**Table 3 materials-14-07114-t003:** Quantitative characteristics of the Mg_2_Si phase formed during sintering at 580 °C.

Material	Average Surface Area (µm^2^)		Area fraction, A_A_ (%)	Dimensionless Shape Factor	Area (µm^2^)	Count	Overall Count
	Value (µm^2^)	Standard Deviation					
Mg-2(n)Si/580	2.53	15.28	9.42 ± 1.53	0.82 ± 0.30	<0.1	2157	12035
					0.1–1	7097	
					1–10	2360	
					10–100	371	
					>100	50	
Mg-4(n)Si/580	4.24	62.98	18.11 ± 6.70	0.76 ± 0.31	<0.1	2016	13572
					0.1–1	8723	
					1–10	2467	
					10–100	292	
					>100	74	
Mg-6(n)Si/580	6.55	69.30	28.84 ± 10.67	0.80 ± 0.32	<0.1	2109	16149
					0.1–1	9181	
					1–10	4133	
					10–100	584	
					>100	142	

**Table 4 materials-14-07114-t004:** Quantitative characteristic of α-Mg-based areas surrounded by multiphase skeleton formed during sintering at 580 °C.

Material	Average Surface Area (µm^2^)		Area fraction A_A_ (%)	Dimensionless Shape Factor	Area (µm^2^)	Count	Overall Count
	Value (µm^2^)	Standard Deviation					
Mg	3812.21	5711.32	95.78 ± 0.39	0.38 ± 0.22	<100	96	585
					100–1000	108	
					>1000	381	
Mg-2(n)Si/580	1189.22	2628.69	79.22 ± 4.32	0.64 ± 0.27	<100	816	1392
					100–1000	240	
					>1000	336	
Mg-4(n)Si/580	637.03	2142.68	63.11 ± 5.70	0.71 ± 0.29	<100	1563	2070
					100–1000	243	
					>1000	264	
Mg-6(n)Si/580	484.91	1623.05	56.29 ± 5.09	0.68 ± 0.28	<100	1824	2276
					100–1000	247	
					>1000	205	

**Table 5 materials-14-07114-t005:** Results of HV3 hardness and porosity measurements for composites sintered at 580 °C.

Property	Mg/580	Mg-2(n)Si/580	Mg -4(n)Si/580	Mg-6(n)Si/580
Hardness HV3	41 ± 1.26	47.7 ± 1.89	56.55 ± 3.25	60.95 ± 2.33
COV (%)	3.06	3.97	5.75	3.82
Open porosity (%)	0.98	0.88	0.59	0.78

**Table 6 materials-14-07114-t006:** Results of HV0.2 microhardness and nanoindentation measurements determined for different regions of composite sinters’ microstructure.

Property	Mg			Mg-2(n)Si/580		
	Primary Grains	Grains Boundaries	Mg_2_Si Particles	Matrix	Mg_2_Si Skeleton	Mg_2_Si Particles
Hardness HV0.2	41.98 ± 2.81	40 ± 3.85	-	56.25 ± 8.84	104.52 ± 65.7	-
COV (%)	6.69	9.64		15.72	62.86	
Nanohardness (GPa)	0.85 ± 0.07	3.85 ± 2.17	-	0.76 ± 0.15	1.97 ± 0.5	7.20 ± 1.86
COV (%)	7.99	56.29		19.7	24.88	25.83
Modulus (GPa)	42.2 ± 4.02	85.93 ± 25.93	-	40.38 ± 3.85	60.1 ± 10.9	128.95 ± 19
COV (%)	9.53	30.17		9.53	18.14	14.73
**Property**	**Mg-4(n)Si/580**			**Mg-6(n)Si/580**		
	**Matrix**	**Mg_2_Si Skeleton**	**Mg_2_Si Particles**	**Matrix**	**Mg_2_Si Skeleton**	**Mg_2_Si Particles**
Hardness HV0.2	60.8 ± 11.15	139.84 ± 94.49	-	61.95 ± 12.78	136.56 ± 77.35	-
COV (%)	18.34	67.57		20.63	56.64	
Nanohardness (GPa)	0.79 ± 0.1	4.14 ± 2.15	3.35 ± 2.56	0.9 ± 0.05	3.1 ± 1.55	6.08 ± 2.1
COV (%)	12.79	51.92	76.52	5.35	50.15	34.54
Modulus (GPa)	47.31 ± 2.92	89.91 ± 21.41	68.65 ± 46.21	45.53 ± 2.9	76.23 ± 13.88	100.84 ± 27.15
COV (%)	6.16	23.81	67.31	6.37	18.21	26.92

**Table 7 materials-14-07114-t007:** Results for measurements of Mg-Mg_2_Si composites sintered at 640 °C.

Material	Hardness HV3 COV (%)	Open Porosity (%)	Mg_2_Si area fraction A_A_ (%)
Mg-2(n)Si/640	138 ± 16	1	40.08 ± 6.51
	11.59		
Mg-4(n)Si/640	149 ± 13	0.53	46.32 ± 17.14
	8.72		
Mg-6(n)Si/640	155 ± 14	0.08	50.9 ± 18.83
	9.03		

## Data Availability

Data is contained within the article.

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
