# Peer review of "Application of Nanosilicon to the Sintering of Mg-Mg2Si Interpenetrating Phases Composite"

_materials, 2021, doi:10.3390/ma14237114_

Round 1
Reviewer 1 Report
Paper is experimentally based. It derves to be published.
Two points can be improved: line 312: it is "Mg2S", it should be: "Mg2Si"
Table 6.: what it means "Hardness 41.98 +-" ? Data with "+-" should be written together in one line.
Author Response
Dear Reviewer,
We would like to kindly thank for your valuable comments.
“Two points can be improved: line 312: it is "Mg2S", it should be: "Mg2Si"
Table 6.: what it means "Hardness 41.98 +-"
Data with "+-" should be written together in one line. “
We improved the points according to your comments.
Yours faithfully,
Authors team
Reviewer 2 Report
The current article discusses the influence of addition of nanosilicon to pure Mg to in-situ develop Mg2Si secondary phase. The topic is worthy of an investigation and the authors have done an excellent job in designing the experiments. I have few suggestions and seek some clarifications from the authors before the manuscript can be considered for publication.
- Some recent articles on powder metallurgy synthesized ceramic reinforced Mg nanocomposites must be added. Few suggestions are below.
SiO2 - https://doi.org/10.1557/jmr.2017.194
Sm2O3 - https://doi.org/10.3390/met7090357 - Line 134-139. There is a possible repetition of the sentence "at least 30 indications...."
- Line 166-177. Traditionally, increase in vol % of nano reinforcements increases the agglomerated sites. In this case, 6 vol % shows the most uniform distribution compared to the other two samples. This observation is contradicting the one presented in the sintered samples. Revise, if necessary for better readability or elaborate.
- Fig 2c displays lower nanoparticle presence than Fig 1c. What is the reason?
- Comment on the wettability between the matrix and the in-situ Mg2Si post mixing and post sintering, respectively.
- The authors discuss DSC results to identify the onset temperature of Mg2Si in both micro and nano sized samples. The results are well explained but what is the significance of this observation. How does lower onset temperature makes in-situ composites more desirable compared to traditional nanocomposites?
- Which sample is represented in Fig 6? It would be better to include all three powder mixture XRD.
- Fig 7c. Ideal to mark with arrows as the color difference cannot be appreciated. Similar to 8c and 9c.
- Fig 10. Which composition is used for the image analysis? Include in the caption. Magnification is provided but scale bar is missing.
- Line 312. Change to Mg2Si.
- What is COF? Introduce the term for better readability.
- Line 347-350. The open porosities and HV3 is not linearly related as claimed by the authors.
Author Response
Dear Reviewer,
We would like to kindly thank for your valuable comments.
Our answers are as follows:
- Some recent articles on powder metallurgy synthesized ceramic reinforced Mg nanocomposites must be added. Few suggestions are below.
SiO2- https://doi.org/10.1557/jmr.2017.194
Sm2O3 - https://doi.org/10.3390/met7090357
ANSWER:
The suggested works for citation are included in bibliography.
- Line 134-139. There is a possible repetition of the sentence "at least 30 indications...."
ANSWER:
We have provided that information for each type of measurement separately, however the number of measurements is same.
- Line 166-177. Traditionally, increase in vol % of nano reinforcements increases the agglomerated sites. In this case, 6 vol % shows the most uniform distribution compared to the other two samples. This observation is contradicting the one presented in the sintered samples. Revise, if necessary for better readability or elaborate.
ANSWER:
We presented in article SEM micrographs of powder mixtures with similar 3 different, and relatively high magnifications, to make possible the comparison of the (n)Si layer deposited on Mg. Therefore, single (n)Si agglomerates not connected with Mg are not visible in Fig. 3 and that information is presented in the text. But please notice the differences in (n)Si arrangement. In Fig. 3 a layer of (n)Si is smooth (3a) and compact (3b and c). It means also that thickness is higher and (n)Si bridges more compacted. At the earlier stages of the (n)Si deposition or for less (n)Si content in initial mixture the layer is thinner and less uniform.
- Fig 2c displays lower nanoparticle presence than Fig 1c. What is the reason?
ANSWER:
It can be an illusion. Fig.1c presents only fragment (secondary agglomerated (n)Si particles from the surface presented in Fig 1b. In the other microareas of the same Mg particle the content of (n)Si is much lower.
- Comment on the wettability between the matrix and the in-situ Mg2Si post mixing and post sintering, respectively.
ANSWER:
We are not sure that we understood correctly that question. It is difficult to discuss about the wettability in our experiments, because silicon transforms to silicide by Mg consumption under pressure. In sintering at 580°C the reactive diffusion is less intense, while at 580°C it is a typical SHS.
- The authors discuss DSC results to identify the onset temperature of Mg2Si in both micro and nano sized samples. The results are well explained but what is the significance of this observation. How does lower onset temperature makes in-situ composites more desirable compared to traditional nanocomposites?
ANSWER:
We presented thermal effects for macrosized and nanosized silicon to show that the application of nanosized componets requires a special attention due to possible differences. In selection of technological parameters that issue is particularly important because of exothermal effects in Mg2Si synthesis. When the (n)Si will be replaced with (n)SiO2 the thermal effects will be few times stronger (see Table 1 in article).
If in the review “traditional nanocomposites” means an application of nano Mg2Si and fabrication of ex situ composite, the DSC results in our article only showed that the change of components size can influence the thermal effects parameters.
- Which sample is represented in Fig 6? It would be better to include all three powder mixture XRD.
ANSWER:
Fig. 6 was changed as suggested.
- Fig 7c. Ideal to mark with arrows as the color difference cannot be appreciated. Similar to 8c and 9c. ?
ANSWER:
We marked structural elements in Fig. 7c as suggested.
- Fig 10. Which composition is used for the image analysis? Include in the caption. Magnification is provided but scale bar is missing.
ANSWER:
All obtained materials were examined by image analysis. In Figure 10 we showed an example of procedure for Mg2(n)Si/580composite.
Figure caption was changed.
- Line 312. Change to Mg2Si
ANSWER:
The editorial error was changed.
- What is COF? Introduce the term for better readability.
ANSWER:
It was our editorial error and was improved, should be COV – Coefficient of Variation
- Line 347-350. The open porosities and HV3 is not linearly related as claimed by the authors.
ANSWER:
In the article we wrote “In general, increasing (n)Si concentration in the initial powder mixture increased the HV3 hardness in Mg-based sintered composites. In the case of Mg-6(n)Si/580, the hardness was 50% higher than the sintered pure Mg, which served as a reference sample. The improvement of hardness was due to silicide formation and reduced porosity compared with sintered pure Mg”. It does not mean linear relation but only properties improvement.
Yours faithfully,
Authors team
Reviewer 3 Report
Authors studied the mechanical property and microstructure of Mg-Mg2Si composite prepared by powder metallurgy. The topic is of interest to the community of Mg alloys. Minor revision, however, is needed.
- XRD patterns of Mg-Mg2Si in Figure 6 are incorrect. There are no 2theta peaks of Mg at 24 and 40 degrees. In addition, 2theta peaks of Mg2Si are also incorrect.
Author Response
Dear Reviewer,
We would like to thank for your valuable comment.
“XRD patterns of Mg-Mg2Si in Figure 6 are incorrect. There are no 2theta peaks of Mg at 24 and 40 degrees. In addition, 2theta peaks of Mg2Si are also incorrect.”
It was our evident editorial fault, we improved.
Yours faithfully,
Authors team
Round 2
Reviewer 2 Report
1. Couple of places in the revised manuscript still mentions COF. Amend accordingly.
Author Response
Dear Reviewer,
errors have been corrected. We are terribly sorry for this oversight.
Best regards in the name of authors